# Pediatric Advance Care Planning: A Scoping Review

**DOI:** 10.3390/children10071179

**Published:** 2023-07-07

**Authors:** Nadine Lusney, Camara van Breemen, Edlyn Lim, Colleen Pawliuk, Zahra Hussein

**Affiliations:** 1Canuck Place Children’s Hospice, Vancouver, BC V6J 2T2, Canada; cvanbreemen@canuckplace.org; 2Department of Pediatrics, University of British Columbia, Vancouver, BC V6H 3V4, Canada; edlyn.lim@bcchr.ca (E.L.); cpawliuk@bcchr.ca (C.P.); 3Child Health British Columbia, Vancouver, BC V6J 4Y6, Canada; zahra.hussein@phsa.ca

**Keywords:** pediatric advance care planning, pediatric palliative care, scoping review

## Abstract

To assess current practice and provide a basis for a provincial template, clinicians at a Canadian pediatric hospice reviewed the literature surrounding pediatric advance care planning (pACP) documentation, process, and implementation for children/youth. The scoping review protocol was developed in accordance with the Joanna Briggs Institute methodology for scoping reviews, and was prospectively registered on the Open Science Framework. MEDLINE, Embase, CINAHL, the Web of Science Core Collection, and Google Scholar, as well as sources of unpublished studies and grey literature, were reviewed. Sixty-four articles met the criteria. Fifteen pACP documentation forms/tools were also identified. Overall, the included articles highlighted that pACP should be a structured, collaborative and iterative process between the family and a trusted or relevant healthcare provider, encompassing medical and non-medical issues. Few articles provided insights into specific elements recommended for advance directive forms. Identified strategies for implementation included a structured, step-by-step pACP interventional tool along with ongoing training, mentorship, and defined organizational structures for the clinician. No single specific ACP tool was acknowledged as the gold standard. Use of a pACP tool, along with ongoing provider education and communication skill development, standardized/accessible documentation, and system-wide quality improvement support, were noted as integral components of pACP.

## 1. Introduction

Currently across Canada, there are inconsistencies in practice regarding overarching provincial guidance and documentation of goals of care (GOC) for children with serious illness medical complexity (i.e., life-threatening/life-limiting illnesses) who require advance care planning (ACP); furthermore, there is a lack of clarity regarding the law and provincial legislature. Pediatric ACP (pACP) is a process that seeks to explore the values, beliefs and wishes to support informed healthcare decision-making and guide future healthcare decisions for a child/family [1,2,3]. pACP recognizes the role, voice and values of the child, and the relationship of the child within a family context. pACP requires a high degree of communication competency, and can lead to better outcomes for families and systems—research has shown that pACP is associated with a decreased number of emergency visits, enhanced ability of parents to plan for most ideal location of care, and decreased parental decisional regret and child suffering at the end of life [4,5,6,7]. Pediatric clinicians face additional barriers to clinicians in adult palliative care that include caring for children too young to express their wishes, collaborating with parents as surrogate decision-makers, treating rare diseases for which prognostication is particularly difficult, and interacting with multiple healthcare providers and teams with differing communication competencies [4,5,6]. ACP can also reduce healthcare provider stress and moral distress by providing clarity regarding family wishes during crises [8]. In pediatrics, this also includes considering the child’s ability to understand, participate and make specific decisions about their care. pACP involves the concept of parallel planning—planning for the life of the child, while also planning for deterioration/death, to allow for the child’s full potential [9]. This primes the mobilization of services and healthcare professionals when necessary. An ACP may, at times, look like an anticipatory care plan—laying out actions to be taken if or when a child’s illness is unstable or deteriorates, or the child develops life-threatening complications due to their illness. Having a specific overarching provincial guideline for initiating and documenting complex conversations surrounding ACP, GOC, and levels of interventions specific to pediatrics, serves to enhance clinicians’ confidence, maintain a consistent message from clinician to clinician, and addresses family’s concerns that may arise [8,10].

Canuck Place Children’s Hospice (CPCH) is the provincial pediatric palliative care program in British Columbia (BC). Pediatric palliative care is a model of care, appropriate at any stage of illness, and can be provided together with disease-directed treatment. It involves integrated care directed at the physical, emotional, spiritual, and social needs of the child and family, delivered by an integrated and multidisciplinary team [11]. CPCH currently supports hundreds of families across BC with co-management of health conditions, pain and symptom control, ACP and enhanced communication, family support and care coordination. The number of children with serious illness/medical complexity in the province who require ACP and documentation of medical intervention is estimated to be >500 each year. Ongoing data analysis suggests that the number may exceed 1000 children, greater than the number of children and families referred to the CPCH program. Documenting GOC occurs in a variety of settings that are often poorly linked, potentially creating tension and difficulty for parents who are attempting to communicate their wishes. Poor coordination of GOC also leads to confusion for healthcare providers about the type of care to provide, as it is unclear as to how or where to document GOC in order to promote continuity of care and shared understanding. Currently, for pediatric patients, there is no provincial document like the BC Medical Orders for Scope of Treatment form (MOST), which is specifically for adult care. The number of children requiring ACP in BC is relatively low in comparison to the adult population; however, the positive impact on the family, healthcare providers, and community could be significant [8,9,10].

Laws and legislature related to pACP differ across and within countries; however, the literature recognizes the importance of integrating pACP into care of children with a serious illness. We sought to review the literature surrounding the pACP process, documenting, and implementation of the documentation processes, with hopes to inform and support content experts across countries to create overarching processes and pediatric specific forms based on the geographical laws specific to them. A preliminary search of PROSPERO, MEDLINE and the Cochrane Database of Systematic Reviews was conducted, and no current or in-progress scoping reviews or systematic reviews on the topic were identified.

The objective was to conduct a scoping review of research literature and grey literature related to the process, documentation and implementation of pACP.

## 2. Materials and Methods

This scoping review was conducted in accordance with the Joanna Briggs Institute methodology for scoping reviews, and was reported using the PRISMA extension for scoping reviews [12,13]. A protocol was co-created by a research librarian, CPCH and Child Health BC, and prospectively registered on the Open Science Framework [14].

### 2.1. pACP Process

#### 2.1.1. Inclusion Criteria

This review considered studies that explored pediatric ACP process, documentation, and the implementation of ACP documentation from all healthcare settings, including in-home care, hospital and community care, in all geographic locations.

Pediatrics refers to the prenatal, neonate, child, and youth population (0–19 years of age), but may also extend into young adulthood in some contexts. These children are defined as at-risk for sudden death and/or expected death prior to reaching adulthood. Consideration was made to include studies that include a mixed population of pediatric participants and young adults, or studies focused on young adults, as the age range for young adult versus pediatric might be different in other countries.

This scoping review included quantitative, qualitative, and mixed methods study designs. In addition, systematic reviews and text and opinion papers were assessed, as were pACP documents, practices, and processes identified in our environmental scan. This review only considered studies published in English, due to a lack of resources to translate non-English articles. Modifications to the inclusion criteria included excluding any that lacked the full report (such as conference abstracts).

#### 2.1.2. Information Sources and Search Strategy

An initial limited search of MEDLINE (Ovid) and CINAHL (EBSCOhost) was undertaken to identify articles on the topic. The text words contained in the titles and abstracts of relevant articles, and the index terms used to describe the articles, were used to develop a full search strategy for MEDLINE (Ovid). The search strategy employed a pediatric search filter developed by the University of Alberta Libraries [15]. The search strategy, including all identified keywords and index terms, was adapted for each included information source as needed.

MEDLINE (Ovid; 1946–2021), Embase (Ovid; 1974–2021), CINAHL (EBSCOhost; 1982–2021), the Web of Science Core Collection (1900–2021), and Google Scholar were searched from inception to 9 October 2021. The search was limited to English, and no date limits were placed on the search.

Sources of unpublished studies and grey literature were searched to find theses and dissertations (ProQuest Theses and Dissertations Global and Networked Digital Library of Theses and Dissertations), conference proceedings (PapersFirst and Proceedings via WorldCat First Search), Canadian organizational websites, and pediatric palliative care programs and hospices in Australia, United States, New Zealand, and United Kingdom (Google searches). See Appendix B for the full search strategies in all databases, and for a complete list of grey literature sources. Experts were contacted for additional grey literature that was not already included in the review.

#### 2.1.3. Study Selection

Following the search, all identified records were collated and uploaded into Covidence, and duplicates were removed. Following a pilot test, titles and abstracts were screened by two independent reviewers (pediatric palliative care specialist clinicians) for assessment against the inclusion criteria for the review. The full text of selected citations was assessed in detail against the inclusion criteria by two independent reviewers. Reasons for exclusion of full-text papers that did not meet the inclusion criteria were recorded and reported in the scoping review. Any disagreements that arose between the reviewers at each stage of the selection process were resolved through discussion.

#### 2.1.4. Data Extraction

Data were extracted from papers included in the scoping review by two independent reviewers using a data extraction tool developed by the reviewers. The minimum data extracted were:Number of participants;Age range;Country;Study design;Person who conducts the ACP conversation (e.g., GP, specialist, paramedic);Characteristics of ACP forms.

The draft data extraction tool was modified and revised as necessary during the process of extracting data from each included paper. No authors of papers were contacted to provide missing or additional data.

## 3. Results

The search resulted in 2327 records, of which 859 were duplicates. A total of 1469 records were screened for eligibility using the title and abstract, and 1323 were excluded. The full texts of the remaining 145 reports were sought, but three articles were unavailable online and difficult to access, and thus were excluded from the review. The remaining 142 full texts were assessed for eligibility, and 79 were excluded. Exclusion reasons included lack of a full study report (conference abstract), not being specific to the pediatric population, study design/outcome did not orientate to inclusion criteria, and studies that did not involve context to pACP process, documentation, or implementation. Sixty-four articles met the criteria mentioned in Section 2.1.1 [8,9,16,17,18,19,20,21,22,23,24,25,26,27,28,29,30,31,32,33,34,35,36,37,38,39,40,41,42,43,44,45,46,47,48,49,50,51,52,53,54,55,56,57,58,59,60,61,62,63,64,65,66,67,68,69,70,71,72,73,74,75]. See Figure 1 for the PRISMA flow diagram of the study selection process.

Forty-eight studies described the pACP process, twenty-eight described documentation practices, and twenty-two described an implementation of a pACP process and/or documentation practice, see Figure 2. Articles came from a variety of regions: the USA (*n* = 24), the UK (*n* = 14), Germany (*n* = 5), Canada (*n* = 5), Australia (*n* = 4), the Netherlands (*n* = 5), Belgium (*n* = 1), Taiwan (*n* = 1), and Australia/Brazil (*n* = 1). Over one-third of the studies/articles originated from the United States, and almost all studies were from the Western world perspective.

A wide range of studies were included: cross-sectional (*n =* 2), non-randomized experimental study (*n =* 2), anthropologic essay (*n =* 1), comparative evaluation (*n =* 1), educational article (*n =* 1), mixed methods (*n =* 4), narrative review article (*n =* 5), position paper (*n =* 1), program development and evaluation (*n =* 1), resource development and pilot (*n =* 2), retrospective chart review (*n =* 5), review article (*n =* 1), mixed methods thesis (*n =* 1), qualitative (*n =* 21), RCT (*n =* 7), systematic review (*n =* 2), and text and opinion (*n =* 4); see Figure 3.

A wide range of studies were included: cross-sectional (*n =* 2), non-randomized experimental study (*n =* 2), anthropologic essay (*n =* 1), comparative evaluation (*n =* 1), educational article (*n =* 1), mixed methods (*n =* 4), narrative review article (*n =* 5), position paper (*n =* 1), program development and evaluation (*n =* 1), resource development and pilot (*n =* 2), retrospective chart review (*n =* 5), review article (*n =* 1), mixed methods thesis (*n =* 1), qualitative (*n =* 21), RCT (*n =* 7), systematic review (*n =* 2), and text and opinion (*n =* 4); see Figure 3. 

Over half of the articles collected included the perspective of the family or the family voice in combination with clinicians (limited child involvement). Of the studies that included participants, the descriptions of participants were as follows: adolescents with cancer (*n =* 4), children/adolescents with life limiting illnesses (LLI)/serious illness (SI) (*n =* 21), adolescents with human immunodeficiency virus (HIV) (*n =* 4), bereaved families (*n =* 3), families supporting children/adolescents with SI (*n =* 14), and clinicians (*n =* 9); see Table 1. Note that majority of studies that included the adolescent population as a descriptor and are depicted in Figure 4 and Table 1, the actual age range was beyond the known boundaries of adolescence. The studies retrieved that included the children/youths/adolescents and young adults (AYA) involved, or parents of a child/youth/AYA range was 0–35 years, with one study regarding Duchenne muscular dystrophy ranging as high as 45 years. A brief description of each article can be found in the Appendix A, along with some age ranges.

### 3.1. pACP Process

pACP is acknowledged in the literature as a valuable and impactful intervention to the care and future care of children and families living with a serious illness, and supportive to the clinicians who care for these populations. An iterative, relational, collaborative, and shared decision-making process for pACP was seen as ideal [8,9,10,17,20,22,37,41,42,43,48,57,58,59]. Involving the child when possible was highlighted as optimal, or at least ensuring the child is the focus of the ACP conversation, basing discussions around values expressed as important to the specific child or the family as a whole [19,20,33,43,48]. The process itself should be individualized to the child/family context, and explore both medical and non-medical aspects of the child’s care and life [10,17,35,41,43,45,47,48,58]. Some studies highlighted supportive aspects of a pACP process from the parents’ perspective, such as written plans being shared, providing enough time when possible for decision-making, and the importance of listening to and understanding families’ perspectives [10,19,35,41,42,43,57]. Within these conversations, the child/family expressed how choices are important, who to involve, and when to discuss it [17,41,42,43,45,68]. These conversations should occur early and routinely, as well as reviewed at key times: when a child is unwell, unscheduled hospital visits, or if death is anticipated [8,10,17,19,28,41,42,43,57].

A conversation guide or pACP interventional tool serves as a helpful framework, providing language and flow to support clinicians in these conversations [24,26,35,50,51,52,53,54,56]. Conversation guides or interventional pACP tools were acknowledged in several studies as being helpful for both families and clinicians, often embodying aspects seen as key to the pACP process. No one pACP guide, tool or intervention was seen as the gold standard; however, it was acknowledged that use of an intervention tool did impact the parents’ perception of the quality of decisions [71]. One study argued that the decision-making process itself can be viewed as a healing ritual to attenuate moral distress around end-of-life decision-making and support the concept of a ‘good parent’ during times of conflict between the healthcare team and families [17]. Ultimately, conversation guides and tools sought to improve collaboration and communication between clinicians and families by providing a structure to conversations, to ensure important topics were addressed, and to allow clinicians to focus on listening.

Clinician-recognized barriers to pACP process across various studies highlighted multidisciplinary team dynamics, knowledge of the patient/family, prognostic uncertainty, timing, culture, fear of causing harm, logistics, organizational structures which are inflexible, and avoidance [8,16,21,26,42]. In contrast, family-/child-recognized barriers in the literature included preferences for receiving information, prognostic awareness, patient involvement, family dynamics, disagreements with professionals, poor communication, relationships with significant power differentials, timing, and avoidance [16,21,40,42].

Clinician-recognized strategies to support pACP practice included a structured, partnered, step-by-step pACP interventional tool, along with ongoing training/education, mentorship, and supportive organizational structures for the clinician to document and engage in practice [19,21]. An example of an organizational structure is a pediatric palliative care program, or access to pediatric palliative care expertise to support and guide conversations [36,65]. Family-/child-recognized strategies included exploring information needs, ongoing conversations, presence of a good summary document that is shared with relevant clinicians, known triggers to initiate pACP conversations, and involvement of pediatric palliative care services [16,31,42,65].

A structured, supportive, collaborative and ongoing/iterative pACP discussion on medical and non-medical issues between the family and a trusted or relevant care provider is ideal, but requires awareness, training, education and ongoing supportive measures (such as PPC involvement). The literature recognizes the practice of pACP requires the marriage of relational communication skills with expert knowledge.

### 3.2. pACP Documentation

Fifteen pACP documentation forms/tools/strategies were identified through various studies, see Table 2. The pACP items ranged from descriptions of pACP tools, to the development of educational booklets, to adaptations of current adult ACP resources, to targeted pACP novel interventions, to cultural adaptations [18,23,24,25,26,29,31,33,34,35,38,45,49,50,51,52,53,56,64,67,69,71,72,73,75]. Four studies utilized the structure of the family-centred ACP, which incorporates the Lyons ACP survey, Respecting Choices, and Five Wishes [23,24,51,52,53]. A variety of pACP tools were explored as interventions with favorable outcomes identified by both clinicians and families; however, there is a lack of consistent patient reported outcomes for any one tool [56]. Several studies adapted, created or explored educational workbooks aimed at families or children, with varying results; the ability to assess workbook impact is difficult, but it is seen as favourable to have an accessible and optional, tailored choice for some families [38,72,73,75]. Determining ideal dissemination of these pACP resources can be difficult.

Many studies focused on adapting pACP tools for specific populations—teens with cancer, adolescents with HIV/AIDS, cultural adaptations, or for a specific country context [48,49,50,51,62,70,71]. Fewer studies were focused on developing novel interventions [18,29,33]. Use of an ACP educational booklet had varied preferences of engagement from families [38], highlighting the importance of understanding a family’s information preferences and child’s individuality for fit of an pACP interventional tool. Limited articles provided insights into the exact variables recommended for advance directive forms. Of the studies that provided insights into variables, it was noted the level of intervention, direction for emergency department healthcare providers, symptom management guidance, preferred location of care, care team member contact information, organ donation, and insights into specific conditions were helpful [21,55,66,69,70].

Documentation was identified as essential to communicating values, beliefs and wishes for future healthcare, with the proviso that it be standardized, written down, and shared amongst key healthcare providers, as well as owned by the caregivers [9,19,21,41,42,45,56]. Clinicians appeared more concerned with documentation aspects, whereas families were concerned with the process surrounding planning [10,40]. Organizational support related to documentation, such as electronic charting and accessibility of documentation, were seen as key. Overall, studies and documents highlighted the wide variation in the availability and nature of formal pACP documents, as well as the varying prevalence of a written plan in place for child deaths.

### 3.3. pACP Process and Documentation Implementation Practices 

The context of implementation practices was explored in consideration of future supports for any advancement or expansion of current pACP programs. Use of a conversation guide, pACP interventional tool, or booklet were seen as helpful when individualized to the family/child needs. Few reported negative outcomes from utilizing guides or pACP interventional tools, with the exception being one study which reported negative impacts to mood, but these are acknowledged to be outweighed by the importance of the conversation, and no associated adverse effects were found [24]; however, other studies show ACP interventions not to impact anxiety or depression scores [52,53,71]. Provider education and communication skills, standardized/structured/accessible documentation, system-wide quality improvement support, and family-centred care are identified as supportive to pACP. Education, along with familiarity, were mentioned most frequently as necessary to support healthcare providers [8,9,42,56]. Education, through workshops and interventional sessions directed at the clinician or patient/participants, were shown to impact confidence in having discussions with patients or loved ones, and overall knowledge of pACP [18,24,42,44]. One study described an educational ACP simulation training, with a reported high satisfaction rate of 90% feeling confidence to hold ACP discussions after participating [44,60]. One study showed confidence was also supported by working in multidisciplinary teams [30]. Education was viewed as favorable and impactful by participants to encourage pACP and enhance understanding and self-confidence [18,24,26,33,44,45,48,56,60]. Similarly, family-centred pediatric advance care planning interventions positively impacted families’ appraisals of their caregiving and made no impact to distress or strain [64]. Education was seen as essential for clinicians utilizing pACP intervention tools or conversation guides to support confidence and integration into practice. One study found that comprehensive implementation plans with a structured ACP intervention, support personnel, learning modules, and organizational/structural clinician supports as facilitators to optimal care of children with medical complexity [33].

## 4. Discussion

Pediatric ACP is becoming more important as the health care landscape changes. More children are living with medical complexity, and technologic interventions are becoming more available to the home setting. All of this means that children are living longer, with more care needs and increasing fragility. Parents, who are providing most of the care, need clinicians that can support planning and medical directives that are aligned with the trajectory of illness, values of the parents, and location of care of the child. Emphasis on the use of a guide or tool to support pACP conversations was clear, as well as the expanding field of pACP [24,26,35,50,51,52,53,54,56]. Education, mentorship and organizational structures were identified as foundational to the process of ACP, normalizing its place in pediatric practice and supporting ongoing practice.

In terms of educating those leading pACP, it is not only important to impress upon the clinicians that pACP is an essential aspect of care, but also the teaching needs to take place in the context of the pediatric population that this region is serving. Educational institutions need to recognize the importance of communication principles. It was recognized in the studies that education was essential [18,24,26,33,44,45,48,56,60], which leads us to believe this is not a skill mastered or supported in medical training or through mentorship at the beginning of one’s careers. The first step is having a systematic approach to identifying children with pACP needs, and creating the space and time in which to provide pACP. Perceived adequate training and settings conducive to these conversations appear to support identification of children with pACP needs to initiate conversations [77]. Clinicians’ perceived triggers to initiate these conversations are primarily deterioration in the health of the child, followed by communication, parent cues, and diagnosis and paediatric intensive care unit admission [77]. Secondly, clinicians need to have an opportunity to practice and be mentored in communication practices and conversation tools that will be family and child centered. For clinicians to make recommendations about advance directives, it is necessary that they first understand the parents/child’s values, hopes, and worries, in the event the child becomes sicker. Clinicians and caregivers need to share and understand each other’s expectations and needs surrounding these conversations, to develop shared goals and a space for ongoing conversations—an opportunity to each benefit from these discussions [78]. The clinician needs to be competent in sharing prognosis in a context of many uncertainties, while remaining truthful, sensitive and informed about communication practices that are helpful to parents. A tool can support their practice; however, training and education still remain essential. This education can come in the form of workshops, simulation practice, experiential learning, and ongoing mentoring by those who are well-skilled in serious illness conversation and care planning. ACP simulation training is both a valuable and feasible educational tool, with positive responses by clinicians and impact to confidence in regard to advance care planning [44,60,79]. Simulation training provides a forum for a clinician to practice a skill and move past the theoretical, with impact from both in person and virtual offerings [79].

Finally, documentation systems and systematic review of pACP goals need to be accessible and agreed upon, and congruent with the area’s laws and legislature. pACP is not a one-way conversation, but rather a process that includes identified triggers (hospital admission, relapse of disease, or parents’ questions regarding new innovative therapy, for example) and response by clinicians. In turn, studying this system through evaluation of parents and clinician experience will only enhance the process and system for future care. Ultimately, the training and education of medical professionals, and the education institutions who house these programs, along with the health authorities, need emphasis and training focused on communication, ACP and current supporting practices. Further research into the apparent gaps and learning needs within educational institutions, residency programs, other allied health and post-school training would be helpful. In addition, ongoing research into understanding a caregiver, child and family perspective on goal setting and ‘regoaling’ would be helpful to support clinicians’ understanding of how to facilitate these conversations [80]. The current research landscape is also sparse for advance care planning with a child who is not a mature minor, as well as the populations experiencing health inequities. Further research is needed to explore the process of advance care planning with school-aged children and promoting agency and involvement in the shared decision-making process. Studies that would include developmentally appropriate and culturally diverse interventions to engage a child in this process, as well as the caregivers’ perspective would be of value. More research is needed in low- and middle-income countries (LMIC) to better understand cultural norms, barriers and opportunities to enhance equity and expand pediatric advance care planning and pediatric palliative care to a global perspective.

Study limitations included limited family participants when the parents’ perspectives were combined with clinician perspectives, and limited inclusion of the perspectives of young children. In particular, studies including adolescent perspectives extended well beyond the adolescent age, and often included ages upwards of 25. Young child and adolescent voices are limited.

## 5. Conclusions

An ongoing, structured, supportive, collaborative and shared pACP discussion is ideal. pACP tools/documents serve as structural supports to the pACP discussion, but require training, education, and ongoing supportive measures for continued confidence and implementation success. Designated supportive clinicians, or the role of pediatric palliative care, appear helpful to support these ongoing conversations. pACP is represented in the literature as impactful to care, valuable to children and families living with a serious illness, and supportive to the clinicians who support and care for these populations. Further research and studies are needed for LMIC to better understand barriers, inequities, and ultimately care for all children globally. Documentation is a requirement of care to ensure care provided to children is in alignment with goals, preferences and wishes. Ensuring pediatric programs align with pACP best practices, integrate supportive structures, and provide ongoing education and training, is paramount to family engagement in serious illness conversations and, ultimately, planning for their child’s future and appropriate use of healthcare resources. Evidence from this scoping review encourages pediatric programs to implement and explore strategies to integrate pACP best practices—including establishing structure, evaluation, training programs, ongoing competency support, exploring inequities, and integrated documentation systems to facilitate support to this intervention.

## Figures and Tables

**Figure 1 children-10-01179-f001:**
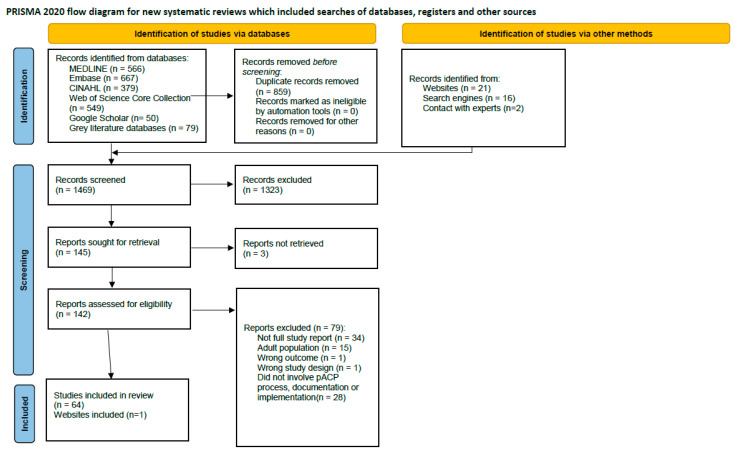
PRISMA flow chart of study selection process [76].

**Figure 2 children-10-01179-f002:**
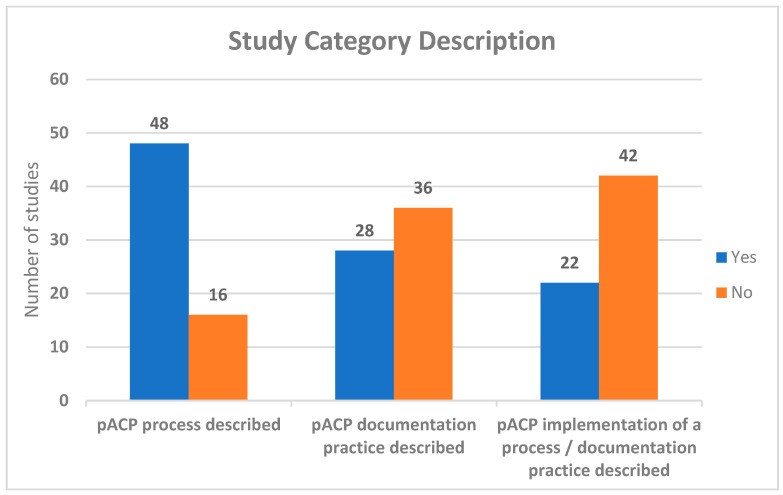
Study category description.

**Figure 3 children-10-01179-f003:**
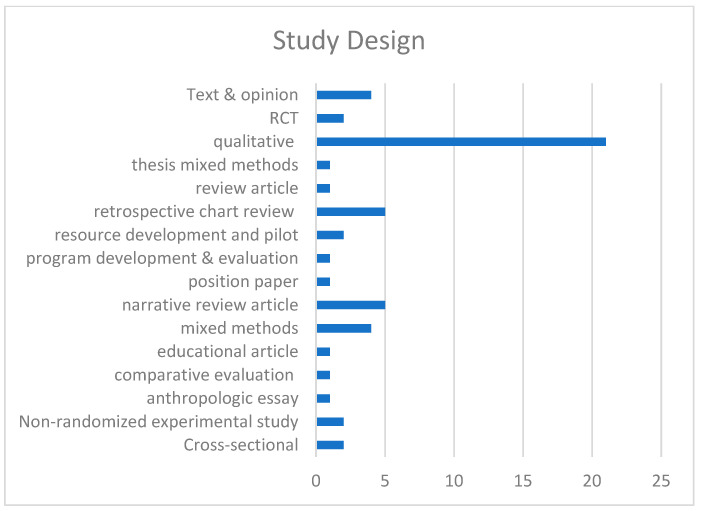
Study Design.

**Figure 4 children-10-01179-f004:**
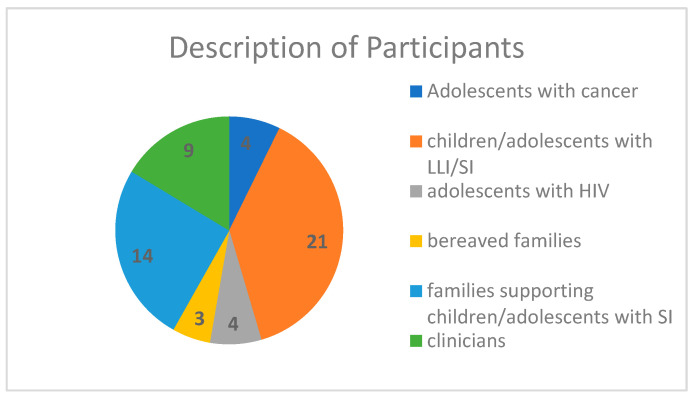
Description of participants in the studies.

**Table 1 children-10-01179-t001:** Data extraction details, description of participants and person conducting ACP.

Description of Participants	Age Range of Participants	Person Conducting ACP
Adolescents with cancer (*n =* 4), children/adolescents with LLI/SI (*n =* 21), adolescents with HIV (*n =* 4), bereaved families (*n =* 3), families supporting children/adolescents with SI (*n =* 14), clinicians (*n =* 9)	0–35 (most articles 0–25, one article mentioned the range up to 45 years of age)	Physician, nurse practitioner, families, allied health, nurse, trainee, trained interviewer

**Table 2 children-10-01179-t002:** Data extraction details.

pACP Documentation Tools/Guides/Strategies
Voicing my Choices, Lyon Advance Care Planning survey, Respecting Choices ACP interview, Five Wishes, My Wishes, Caring Decisions handbook, Statement of treatment preferences, My Choices booklet, Family Centered Pediatric ACP for teens with cancer (FACE-TC), BOOST pACP summary sheet, “My thoughts, my wishes, my voices” document, POLST form, free text in progress notes, Emergency care plans, DNR forms, Pediatric Serious Illness Conversation Guide

* Including the asterix symbol retrieves every potential suffix variations of the word.

## Data Availability

The data presented in this study are openly available in Open Science Framework at https://doi.org/10.17605/OSF.IO/E6K8C (accessed on 30 May 2023).

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
