# Peer review of "Pediatric Advance Care Planning: A Scoping Review"

_children, 2023, doi:10.3390/children10071179_

Round 1

Reviewer 1 Report

Congrats, the scoping study conducted in this research provides valuable insights into the current state of knowledge on pediatric advance care planning (pACP), highlighting key themes and research gaps in the field.

While the review offers a comprehensive overview of the literature, there are a few aspects that could be improved:

•           Inclusion criteria: The review mentions that 64 articles met the criteria but did not specify the inclusion criteria used to select those articles. Providing information on the criteria used to determine article eligibility would enhance the transparency and replicability of the review process.

•           Recommendations for future research: The abstract could benefit from a section that outlines recommendations for future research in pediatric advance care planning. This would provide a clear direction for researchers and highlight areas for further investigation.

Introduction:

It highlights the importance of pACP in supporting informed healthcare decision-making and guiding future healthcare decisions for children with serious illness medical complexity. The introduction also emphasizes the need for clarity regarding the law and provincial legislature related to pACP. The introduction sets the stage for the scoping review by addressing the current gaps and challenges in pACP practice.

Materials and Methods section

The review follows the Joanna Briggs Institute methodology for scoping reviews, a suitable approach for addressing the research objectives.

The section details the inclusion criteria, information sources, search strategy, study selection process, and data extraction. The review utilized multiple databases and sources to search relevant articles comprehensively. The screening and selection process involved independent reviewers, enhancing the reliability of the study selection. The data extraction process is also clearly described, including the specific data elements extracted from each paper.

Overall, the Materials and Methods section demonstrates a well-defined research design and provides sufficient details to understand the methods employed in the scoping review.

Results

In the section describing the results of the study selection process, it would be helpful to provide a brief explanation or clarification for the specific exclusions of the 79 articles that were excluded after full-text assessment.

It would be beneficial to include a summary of the main findings related to barriers and challenges identified in the pACP process, both from the clinician and family/child perspectives. This would provide readers with a concise overview of the factors that may impede the implementation of pACP and facilitate a better understanding of the study's implications.

Discussion/conclusion

The discussion and conclusions of the study seem to be supported by the results. The discussion highlights the importance of pACP in the changing healthcare landscape, where more children live with complex medical needs and receive technological interventions at home. The expansion of pACP and the need for education, mentorship, and organizational structures to support its implementation are emphasized.

The discussion highlights the significance of education and training for medical professionals and educational institutions in communication, ACP, and current supporting practices.

The limitations of the study are also mentioned, including limited family and young child perspectives in the analysis.

Overall, the discussion and conclusions appear well-supported by the study's results.

For your consideration - Specific Recommendations on the conclusion: It could provide more specific recommendations for action based on the study's findings. For example, it could suggest strategies for integrating pACP best practices into pediatric programs, establishing training programs for clinicians, or implementing documentation systems that facilitate ongoing conversations and review of goals.

Future Research Directions: The conclusion could also mention potential areas for future research. This could include exploring the perspectives of caregivers, children, and families on goal setting and adjusting goals over time, identifying gaps and learning needs in educational institutions and residency programs, or evaluating the effectiveness of different educational approaches and interventions.

Author Response

Thank you for your review. Here are my responses:

Inclusion criteria: inclusion criteria is outlined in section 2.1.1. I have also added a highlighted section to the manuscript under the results to refer back to that section to support the reader.

Recommendations for future research: I have added recommendations for future research to the manuscript. Highlighted area in manuscript: The current research landscape is also sparse for advance care planning with a child who is not a mature minor, as well as populations experiencing health inequities. Further research is needed to explore the process of advance care planning with school aged children and promoting agency and involvement in the shared decision making process. Studies that include developmentally appropriate and culturally diverse interventions to engage a child in this process, as well as the caregivers' perspective would be of value. 

Results: A description was added to the results section to provide context as to why 79 articles were excluded. Added wording (highlighted in manuscript): Exclusive reasons included lack of a full report (conference listing), not specific to the pediatric population, study design/outcome did not orientate to inclusion criteria and studies that did not involve context to the pACP process, documentation or implementation. 

Reviewer suggested a summary of the main findings related to barriers and challenges identified in the pACP process, both from the clinician and family/child perspectives. A summary of these findings can be found on lines 218-224. As well as a summary of recognized strategies on lines 225-233.

Discussion/conclusion: See comment above for additions to recommendations for future research. Recommendations to education were mentioned already on lines 349-353. 

Conclusion: More specific recommendations are made in the last sentence of the scoping review. It reads: Evidence from this scoping review encourages pediatric programs to implement and explore strategies to integrate pACP best practices -- including establishing structure,  evaluation, training programs, ongoing competency support and integrated documentation systems to facilitation support for this intervention. 

Reviewer 2 Report

The topic of the work is current and important. This scope review, based on a systematic review, demonstrates the importance of planned pediatric care for children with chronic, serious illnesses, often with a poor prognosis. I congratulate the Authors on the very good preparation of the article and fully recommend it for publication.

Summary: It introduces the subject of the article well and exhaustively. It was explained that Canadian Children's Hospice clinicians reviewed the literature on the documentation, process and implementation of pediatric care planning (pACP) for children/adolescents. A literature review methodology is presented. Eligible articles (64) and 15 pACP documentation forms/tools were indicated. The conclusions provided integral elements of pACP, which are based on the use of pACP tools, along with continuous service provider education and communication skills development, standardized documentation, and system-wide support for quality.

Keywords: Scope is valid and applies to all items analyzed in the scope overview.

Introduction: The authors explain in detail the need to review and develop care plan documentation for children, as does BC Medical Orders for Scope of Treatment (MOST) Form 71, which is specifically for adult care. They explain that there are currently inconsistencies in practice in Canada regarding guidance, documentation and legislation for the care of children with serious life-threatening illnesses. Meanwhile, pediatric palliative care is a model of care that is appropriate at each stage of the disease and includes integrated care focused on the physical, emotional, spiritual and social needs of the child and family, delivered by an integrated and multidisciplinary team.

Therefore, the purpose of this paper is to review the scope of the scientific and grey literature related to the process, documentation and implementation of pACP due to the lack of current or pending reviews on this topic.

There is an unnecessary space on line 80.

Materials and Methods: This scope review was conducted in accordance with the Joanna Briggs Institute scoping review methodology using the PRISMA extension. Inclusion and exclusion criteria have been developed. MEDLINE (Ovid; 1946-2021), Embase (Ovid; 1974-2021), CINAHL (EBSCOhost; 116 1982-2021), Web of Science Core Collection (1900-2021) and Google Scholar were searched from inception until October 9, 2021. Ultimately, 64 studies met the inclusion criteria. The protocol is prospectively registered with the Open Science Framework.

Results: Figure 1 provides a flowchart of the PRISMA study. Based on the literature, the pACP process, documentation and principles of implementing pACP into practice were discussed. A detailed supplement to the results is included in the attached appendix, which contains data on 73 bibliographic items.

The discussion highlighted the importance of education, mentoring and organizational structure as fundamental to the pACP implementation process. Limitations of the study were also given due to the small number of family participants with children.

Conclusions: It was emphasized that pACP documentation is a requirement to ensure care provided to children in accordance with goals, preferences and wishes.

Author Response

Thank you for your time and review of this article. I have removed the unnecessary space on line 80

Reviewer 3 Report

Well written manuscript.

It appears that the total number of articles analysed is a small fraction of the number of articles searched for - 64/1469 = a mere 4.45%.

It is not clear from the PRISM table as to why the rest of the articles were rejected. What inclusion or exclusion criteria did they failed to satisfy?

In your supplementary material, some of the 'findings' reported are rather unsavoury and perhaps racially prejudiced - for example:  'Previous AD knowledge was significantly more likely among parents and caregivers with high educational degrees, parents/caregivers that were English speaking.' 

It might be pertinent to include any linguistic, socio-cultural or religious reasons that might impact on the very need for so called specialised paediatric palliative care. 

It is essentially a summary of studies published from Caucasian oriented or Western oriented authors. 

While the article is all about paediatric palliative care and advance - directives, the ages included in the articles range as far as 35 years, while it is 25 years in most of the studies. 

Keeping in view the age group mentioned above, the pie diagram mentioning adolescent appears to be incorrect. 

Decision making in palliative care evening the adult setting involves elders in the family. Hence, decision making by parents for their children is by no means a surprise.

The laws of each country are entirely upto their legislating bodies - you could, perhaps site the law as it exists in Canada in this respect. And, add your comments.

An o has been added to 500 while citing the incidence of families seeking paediatric palliative care in British Columbia - it may just be a typographic error. 

It is not clear from the article whether you desire that the objectives of paediatric palliative care are uniform for Canada, North America, Caucasian world or the whole wold.

It would be nice if you can summarise with a few suggestions of your own for the reader - both on 'law' as well as 'practice' of paediatric palliative care and end of life decisions, including advanced directives. 

Author Response

Thank you for your time and review of this article. Here are my responses to your comments and suggestions.

The 4.45% : this was the author's first time completing a scoping review. The two content experts were clear what they were looking for, but the search criteria provided to the research library through the protocol was very broad; thus, more articles were retrieved that were not relevant. For example, many articles were about adults and not specific to pediatric or focused on the "why" ACP is important. 

Clarity with PRISM table: Inclusion criteria is outlined in section 2.1.1. This was the criteria utilized to assess the articles. Expansion on exclusion criteria is added and available on lines 155-159. 

Caucasian oriented/Western oriented: I agree with your comments and agree there is research bias to the Western oriented world.  No inclusion criteria or exclusion was specific to articles being caucasian oriented or western oriented, but I recognize only looking at english written articles could limit. I am not aware of the race or ethnicity of the authors of the articles and studies retrieved and would not like to make assumptions based on individual last names. There were several articles not western related included in the retrieval, but I agree the vast majority of the available research is from the western world and not low middle income countries (LMIC). I have made more explicit recommendations in the discussion and conclusion section regarding this and I thank you for your comments.

Supplemental material: The findings were pulled directly from each study/article. I couldn't locate the description you provided in your comments of 'Previous AD knowledge was significantly more likely among parents and caregivers with high educational degrees, parents/caregivers that were English speaking'. This was a finding by that study, not this author's findings or description. That being said, I appreciate your comments and will review the supplemental material with an awareness of how findings are represented by those study authors. 

Age range: That is correct, there was only one study that had a range to 35 and majority of the articles ranged only up to 25. The authors included any studies/articles that had children involved who were 19 years of age (for example if a study included youth aged 17-25, then we included the article). 

Pie chart: reviewer stated the pie chart appeared incorrect that mentioned adolescents. The pie chart represents any articles that included the mention of a child/adolescent or family, not all the studies included a child/adolescent or family, so the pie chart is only representative of the studies/articles that involved these individuals. Age range was defined earlier in the article and description of the articles retrieved. Can you please expand a bit further so I can understand the inaccuracy you are seeing? 

Typographic error: thank you, I have removed the extra o following 500. 

Suggestions on 'law' as well as 'practice': author has made additional recommendations for research and more explicit wording around the conclusions for suggestions for practice, see highlighted added areas. The authors are not experts in the law, so I am unsure at this time what my recommendations to the law would be. 

Law citing: Authors are aware of the laws specific to British Columbia (ex: Infant Act of BC), but do not feel they are experts in the law and can comment. I have added some recommendations to practice to ensure clinicians are aware of the legislature and law surrounding their practice. 

Round 2

Reviewer 3 Report

Thank you for addressing the reviewers comments.

Adolescents is a defined age group. Your explanation that if the age range covered or included the adolescent age group, it was included in the analysis and hte pie chart would be rather incorrect, the study quoted may have included subjects who were from the intraauterine to nearly in their nineties! 

Hence, to include an article into your analysis should perhaps mention that the quoted study had a substantial number of 'adolescents' in their analysis.

Althouogh you have selected only English language articles, your statement that you would not know the race or ethnicity of the reported study subjects from the last names of the authors is not correct: the article would give details of the geographic region in the world from where the study was reported, albeit the names of the authors might leave you clueless.

Author Response

Thank you for clarity provided. The age ranged mentioned from the retrieved studies is present in both Table 1 and line 182. I have added additional context (lines 182-187) to provide the reader with a more full picture of the literature. I also added elaboration to limitations in lines 369-371 regarding perspective of the young child and adolescent. You are correct, there is very limited perspective from the young child/adolescent in the literature. While some studies included their voice, they were often analyzed alongside "adolescents" upwards of 25 years of age with two studies with higher ranges.

I added to a sentence on line 168 to highlight to the reader the limitation of perspective beyond the Western world.